# Treatment Options for Hepatitis A and E: A Non-Systematic Review

**DOI:** 10.3390/v15051080

**Published:** 2023-04-28

**Authors:** Filippo Gabrielli, Francesco Alberti, Cristina Russo, Carmela Cursaro, Hajrie Seferi, Marzia Margotti, Pietro Andreone

**Affiliations:** 1Postgraduate School of Internal Medicine, University of Modena and Reggio Emilia, 41126 Modena, Italy; 2Department of Surgical Sciences, University of Bologna, 40126 Bologna, Italy; 3Internal and Metabolic Medicine, Department of Medical and Surgical Sciences, Maternal-Infantile and Adult, AOU di Modena, University of Modena and Reggio Emilia, 41126 Modena, Italy; 4Division of Internal Medicine, Department of Medical and Surgical Sciences, Maternal-Infantile and Adult, University of Modena and Reggio Emilia, 41126 Modena, Italy; 5Postgraduate School of Allergology and Clinical Immunology, University of Modena and Reggio Emilia, 41126 Modena, Italy

**Keywords:** hepatitis A virus, HAV, hepatitis E virus, HEV, treatment, antiviral therapy, ribavirin, vaccines

## Abstract

Hepatitis A and hepatitis E are relatively common causes of liver disease. Both viruses are mainly transmitted through the faecal–oral route and, consequently, most outbreaks occur in countries with poor sanitation. An important role of the immune response as the driver of liver injury is also shared by the two pathogens. For both the hepatitis A (HAV) and hepatitis E (HEV) viruses, the clinical manifestations of infection mainly consist of an acute disease with mild liver injury, which results in clinical and laboratory alterations that are self-limiting in most cases. However, severe acute disease or chronic, long-lasting manifestations may occur in vulnerable patients, such as pregnant women, immunocompromised individuals or those with pre-existing liver disease. Specifically, HAV infection rarely results in fulminant hepatitis, prolonged cholestasis, relapsing hepatitis and possibly autoimmune hepatitis triggered by the viral infection. Less common manifestations of HEV include extrahepatic disease, acute liver failure and chronic HEV infection with persistent viraemia. In this paper, we conduct a non-systematic review of the available literature to provide a comprehensive understanding of the state of the art. Treatment mainly consists of supportive measures, while the available evidence for aetiological treatment and additional agents in severe disease is limited in quantity and quality. However, several therapeutic approaches have been attempted: for HAV infection, corticosteroid therapy has shown outcome improvement, and molecules, such as AZD 1480, zinc chloride and heme oxygenase-1, have demonstrated a reduction in viral replication in vitro. As for HEV infection, therapeutic options mainly rely on the use of ribavirin, and some studies utilising pegylated interferon-alpha have shown conflicting results. While a vaccine for HAV is already available and has led to a significant reduction in the prevalence of the disease, several vaccines for HEV are currently being developed, with some already available in China, showing promising results.

## 1. Introduction

Annually, the hepatitis A (HAV) and hepatitis E (HEV) viruses still affect millions of people worldwide. Although these two infections are more common in low-income countries, outbreaks are still detected in high-income countries at present, placing the focus, once again, on effective strategies to prevent and control the infections. Moreover, despite the fact that most infections are asymptomatic or paucisymptomatic, a variable amount of these infections evolves to a severe form, such as acute hepatitis or acute liver failure (ALF), resulting in hospitalisation, the need for a liver transplant or death. At present, no aetiological cure is available, but some efforts have been made to find drugs that are able to alter disease progression. Although HAV and HEV share some common aspects, such as being RNA viruses with an icosahedral non-enveloped capsid and having similar symptoms and transmission modes, both pose unique challenges to public hygiene due to their distinct differences. Figure 1 illustrates some of the characteristic features of these viruses. In this paper, we conduct a non-systematic review of the literature focusing on the epidemiology, pathophysiology and treatments of HAV and HEV infections in different settings. Our aim is to highlight the evidence present in the literature on the different therapeutic options available to treat HAV and HEV infections.

## 2. Materials and Methods

We conducted a non-systematic review using PRISMA guidelines in the following electronic sources: PubMed, Scopus, Google Scholar and ClinicalTrial.gov.

For the hepatitis A virus, we used the following words in our search: (“HAV”) AND (“HEPATITIS A VIRUS”) AND (TREATMENT OR THERAPY). We included free full text, full texts; case reports; classical articles; clinical studies; clinical trials; clinical trial protocols; clinical trials, phase I; clinical trials, phase II; clinical trials, phase III; clinical trials, phase IV; comparative studies; controlled clinical trials; meta-analyses; multicentre studies; observational studies; practice guidelines; randomised controlled trials; and reviews and systematic reviews, published from 1980 to June 2022. Human and in vitro studies were included, and we highlighted data about mortality, survival, need for a liver transplant, sustained virological response, biochemistry values and viral load, wherever possible. We excluded articles not in the English language and pre-prints.

For the hepatitis E virus, we used the following words in our search: (“HEV”) AND (“hepatitis e virus”) AND (“treatment” OR “therapy”). We included free full texts; full texts; case reports; classical articles; clinical studies; clinical trials; clinical trial protocols; clinical trials, phase I; clinical trials, phase II; clinical trials, phase III; clinical trials phase IV; comparative studies; controlled clinical trials; English abstracts; meta-analyses; multicentre studies; observational studies; practice guidelines; randomised controlled trials; and reviews and systematic reviews, published from 1980 to June 2022. Human and in vitro studies were included, and we highlighted data about mortality, survival, need for liver transplant, sustained virological response, biochemistry values and viral load, wherever possible. We excluded articles not in the English language and pre-prints.

We also included data from the main regional epidemiological societies (Center for Disease Control and Prevention, European Centre for Disease Control and Istituto Superiore di Sanità), as well as the WHO “Epidemiology and Prevention of Vaccine-Preventable Diseases” manual. The search was conducted as follows: Dr. Gabrielli (F.G.) and Dr. Alberti (F.A.) identified relevant studies by reading the abstracts and searching for additional studies through the reference lists of the selected papers. Then, Dr. Gabrielli (F.G.) and Dr. Alberti (F.A.) independently reviewed the studies by checking the titles and abstracts of the articles and deciding whether to include each article. Non-original articles and off-topic articles were excluded.

### Article Screening and Selection

In the first step, two reviewers (F.G. and F.A.) independently evaluated the eligibility of all of the titles and abstracts. Studies were included in the full-text screening if either reviewer identified the study as potentially eligible or if the abstract and title did not include sufficient information for exclusion. Studies were also eligible for full-text screening if they included the data on treatment, dosage, virological and biochemical response, and presence of a control group, whenever possible. According to the previously defined inclusion and exclusion criteria, in the second step, the same reviewers independently performed a full-text screening to select articles for qualitative synthesis. Disagreements were resolved by consensus (F.G. and F.A.) or arbitration (P.A.).

A total of 318 articles were found for HAV and 982 for HEV. Flow diagrams regarding the selection of articles for HAV and HEV are shown below (Figure 2 and Figure 3). Considering that almost all studies are case reports or case series, we did not perform an evaluation of risk of bias.

## 3. HAV Infection Overview

The history of hepatitis is probably as long as human existence on the planet. The first records are generally attributed to Hippocrates; however, an illness resembling hepatitis was described in China around 5000 years ago. A more accurate description of hepatitis with characteristics suggestive of hepatitis A virus (HAV) infection was only made in the last three centuries, particularly during the World Wars, when outbreaks of jaundice were referred to as “jaunisse des camps”, literally translating to “jaundice of the battlefields”. The virus responsible for the disease was eventually identified in the early 1970s by Feinstone and his group [2]. In the following years, a serological test was developed, and an inactivated vaccine became available in 1991. As a consequence, HAV infection rates decreased significantly in high-income countries, where it once was the leading cause of infectious hepatitis [3]. HAV is a positive-sense RNA, non-enveloped virus belonging to the family of *Picornaviridae*, *Hepatovirus* genus, of which humans and primates are the only natural host. To date, five genotypes are known, of which only three (I, II and III, each subgenotype in A and B) are infectious to humans [4,5]. Infectious viral particles are intrinsically resistant to adverse environmental conditions, including acidic pH, detergents and drying [6,7,8]. Transmission mainly occurs through the faecal–oral route involving contaminated food or water, although occasionally transmission through sexual practices has been reported. Parenteral transmission due to viraemic phases has also been reported with blood products and potentially in injecting drug users [9,10]. The infection is more common in low-income countries where it is considered endemic, due to poor hygienic conditions, but outbreaks related to contaminated food or water also occur in high-income countries. Cases in developed countries typically affect people considered at high risk, such as travellers, injection drug users, men who have sex with men, isolated populations, sewage workers and homeless individuals [8,11]. If HAV enters an environment, there is a significant risk of outbreaks occurring [8]. A risk factor is identified in around 40% of infected individuals in the USA [11].

### 3.1. Epidemiology

Globally, the WHO estimates 1.5 million cases of HAV per year, which resulted in 7134 reported deaths in 2016. In the USA, the CDC reported 12,474 cases of acute hepatitis A, corresponding to an overall incidence rate of 3.8 cases per 100,000 [11,12]. Yearly, confirmed cases in Europe amounted to 12,429 in 2016, corresponding to an incidence of 2.4 cases per 100,000 population per year. Due to the majority of cases affecting young patients with limited clinical manifestation, the disease may be under reported [13]. In Italy, the Istituto Superiore di Sanità (ISS) reported 126 new cases in 2021, with little increase compared to 2020 [14]. Different outbreaks caused by food contaminated with HAV have been reported in developed countries [15,16,17,18,19]. For example, 1589 people were infected with HAV in Europe due to frozen berries originating from Bulgaria and Poland [15].

### 3.2. Pathophysiology

Once HAV is ingested, it survives the acidic environment of the stomach and reaches the liver. Replication in the gastrointestinal tract is uncertain. Once the liver is infected, viral elimination occurs through the bile into faeces, although it is partially subjected to entero-hepatic recirculation. The usual incubation period for HAV is between 14 and 28 days [7]. Viral replication occurs in the hepatocyte cytoplasm, and cellular destruction results from an immune-mediated mechanism involving CD8-positive T lymphocytes and natural killer (NK) cells against HAV antigens, rather than the direct cytopathogenic effect of the virus. The release of interferon-gamma (INF-γ) from the involved lymphocytes contributes to viral clearance [20]. It was demonstrated that HAV can modulate type I interferon release, causing a blunted response in infected chimpanzees, especially in the first phase [21,22]. These findings could be important to understand why some patients develop relapsing hepatitis [23,24,25]. In the wide spectrum of the severity of the clinical presentation of HAV infection, severe disease with fulminant hepatitis has been associated with a marked reduction in circulating HAV RNA, suggesting that the adverse course may be the consequence of an excessive host response [26]. When host factors were studied, it was found that some subjects with fulminant hepatitis exhibit higher interleukin-18 (IL-18) levels in hepatocytes and macrophages, thus leading to an excessive NK cells response [27]. T-cell immunoglobulin and mucin domain 1 (TIM-1), a cellular receptor for HAV, was also investigated, leading to the conclusion that the severity of hepatitis A could be due to variations in the TIM-1 domain, which can activate NK cells to different degrees [28,29]. Further studies are needed to understand the exact pathophysiology of HAV infection.

### 3.3. Clinical and Laboratory Manifestations

The clinical manifestations of HAV infection range from asymptomatic infection to fulminant hepatitis with acute liver failure and the need for liver transplantation. Age affects disease manifestation, as children are more frequently asymptomatic and only develop clinical disease in around 30% of cases compared to over 70% of adults [30]. Symptom onset is typically abrupt. Fever, malaise, anorexia, nausea, vomiting, abdominal discomfort and headache are the first non-specific, prodromal symptoms to appear. They are followed, over the course of days to weeks, by the signs and symptoms of cholestasis, which include dark urine, pale stools, jaundice, pruritus and hepatomegaly. Prodromic symptoms typically improve by this time, while jaundice peaks at around two weeks from the onset of symptoms. Extrahepatic manifestations, such as skin rash and arthralgia, are frequent, while other uncommon findings include leukocytoclastic vasculitis, myocarditis, cryoglobulinemia, aplastic anaemia, thrombocytopenia and pure red cell aplasia [31,32,33,34].

Laboratory abnormalities include increased aminotransferases, bilirubin and alkaline phosphatase. The increase in serum aminotransferases typically precedes the increase in bilirubin, and it is frequently above 1000 IU/L. Alanine aminotransferase (ALT) is proportionally more elevated than aspartate aminotransferase (AST). Increased bilirubin follows the rise in aminotransferases, is typically within 10 mg/dL and decreases within two weeks of peaking. Non-specific acute phase reactants may also be increased [31,35]. Hepatic complications include fulminant hepatitis, chronic cholestasis, relapsing hepatitis and autoimmune hepatitis. Typically, fulminant hepatitis due to HAV occurs in less than 1% of patients and has a spontaneous survival rate of 70%, while the other 30% either requires orthotopic liver transplantation (OLT) or results in death [36]. Fulminant hepatitis is often associated with advanced age, pre-existing liver injury (such as non-alcoholic liver disease (NAFLD) or alcoholic liver steatohepatitis (ASH)) and host response, as described previously [37]. Prolonged cholestasis is also a rare complication that can occur after HAV hepatitis and possibly results from the interaction between cellular and humoral immunity with pro-cholestatic polymorphisms [38]. Generally, prolonged cholestasis resolves spontaneously, although in some cases, the associated symptoms, mainly itching and malabsorption, require treatment. Up to 20% of HAV infections evolve in relapsing hepatitis. In this subset of cases, the initial manifestations resolve spontaneously with the normalisation of clinical and laboratory findings, followed by a relapse typically within 6 months of the infection. Relapses are usually manifested as laboratory abnormalities (increased aminotransferases above 1000 IU/L, persistence of serum HAV IgM and HAV RNA in stools) [39]. Commonly, relapsing hepatitis is milder than acute hepatitis [39]. Cases of autoimmune hepatitis (AIH) triggered by HAV infection have been reported [40,41,42,43,44].

### 3.4. HAV Treatment

As mentioned before, most cases of HAV are self-limited and treatment is limited to providing adequate supportive care. No specific treatment is available for HAV infection, and support therapy is necessary for symptomatic cases. Patients with fulminant hepatic failure need to be referred to a liver transplant centre. The prevention of infection is the most important action, and it can be achieved in different situations. Hygiene dispositions to avoid infection, such as washing hands, cooking food, and using potable water, are mandatory. Vaccinations of high-risk categories should be performed [45]. Although no specific medication is approved for the treatment of HAV infections, some therapeutic agents have been used experimentally in complicated cases (Table 1). Currently, none of these are licensed for hepatitis A treatment.

#### 3.4.1. Supportive Treatment

Traditional supportive care involves the use of nutritional support, hydration, the use of antiemetics in the case of vomiting and the use of antipyretics to relieve fever. Attention must be paid to the use of paracetamol due to liver toxicity.

#### 3.4.2. Vaccines

Two types of vaccines are currently available: an inactivated one developed in the USA and Europe, and a live, attenuated one used only in China, Bangladesh, Guatemala, Philippines, Thailand and India [46]. These vaccines are indicated for outbreak control and prophylaxis for high-risk subjects, such as travellers and laboratory workers, or high-risk populations. Furthermore, the CDC (Center for Disease Control and Prevention) recommends vaccination in children between 12 and 23 months with a second dose at least 6 months after the first one or in individuals between 2 years old and 18 years old who have not received vaccination, are illicit drug users, people experiencing homelessness, people with chronic liver disease, HIV-positive people and pregnant women at high risk for severe hepatitis A infection. Additionally, HAV vaccines are indicated as post-exposure prophylaxis [47]. Inactivated vaccines are administered by intramuscular injection in two separate doses. Seroconversion is achieved in 95% of individuals after the first dose and in 100% after the second dose [48]. The production of anti-HAV IgM antibodies is observed around a week after vaccination, and they significantly increase the immune response in the case of exposure [49]. On the other hand, anti-HAV IgG is detectable after 1 month from the vaccination. Immune dysfunction due to HIV or leukaemia may result in an impaired response to vaccination [50]. Two different studies have evaluated the efficacy of vaccination versus passive immunisation with immune globulin after exposure, and both concluded that subjects who received HAV vaccines had a higher probability of seroconversion compared to immune globulin alone. However, the simultaneous administration of a vaccine to achieve active immunisation and Ig to obtain passive immunisation may be a reasonable approach to offer adequate protection after HAV exposure [51,52].

#### 3.4.3. N-Acetylcysteine

Gunduz et al. [53] investigated the effect of N-acetylcysteine (NAC) on acute viral hepatitis caused by HAV and HBV. NAC acts by directly reducing oxidative stress, as well as restoring endogenous antioxidative systems, particularly the glutathione one. Administrations of 600 mg per day of NAC did not demonstrate any advantages over a placebo.

#### 3.4.4. Interferon

Few studies investigated the utility of an interferon-based regimen in HAV infection. A study conducted by Crance et al. demonstrated the in vitro inhibition of viral replication [54] due to interferon-alpha regimens. Three patients with fulminant hepatitis A and one patient with an acute severe form were treated with interferon-beta, resulting in the improvement of liver function and survival [55]. Other interferons, such as interferon type III and interleukin-29, have been studied in vitro with interesting results [56].

#### 3.4.5. Sofosbuvir

Sofosbuvir is a drug approved for the treatment of chronic HCV. The antiviral activity is achieved through the inhibition of viral RNA synthesis acting on the RNA-dependent RNA polymerase (RdRp) [57]. Sofosbuvir has been shown to decrease HAV RNA levels in vitro in infected cells without detectable cell toxicity [58].

#### 3.4.6. Corticosteroids

Corticosteroids have been investigated as a complementary therapy of acute HAV hepatitis, based on the rationale that a significant amount of liver tissue injury is immune-mediated. In a study conducted by Zakaria et al. [59], corticosteroids were administered to 18 of 33 children with fulminant hepatitis A. Those who were treated with corticosteroids received either prednisolone at a dose of 1 mg/kg/day or methylprednisolone at a dose of 0.8 mg/kg/day, while the other 15 patients were treated with standard supportive measures. In the corticosteroid treatment group, only 3 deaths occurred compared to 11 deaths in the placebo group, resulting in a statistically significant difference (*p* = 0.001). Different case reports showed an improvement in the disease after corticosteroid administration [60,61,62,63,64,65,66,67,68]. The utility of corticosteroids to prevent prolonged cholestasis after acute hepatitis A has been evaluated in several case reports and case series, where it has been reported a reduction in bilirubin levels, itching and liver injury markers [69,70]. Prednisone is usually started at a dose of 30–50 mg daily and tapered off at a rate of 5 mg every week. Corticosteroids have also been used in pure red cell aplasia, a very rare complication of HAV.

#### 3.4.7. Liver Transplantation

One-third of patients with acute liver failure caused by HAV need a liver transplant [68]. Transplantation in HAV infection has lower survival rates compared to HBV: at 1 year, 69% vs. 88%, respectively [71]. A retrospective review conducted by Navarro et al. on liver transplant evolution for fulminant liver failure due to HAV suggested that particular attention is warranted in patients with pre-existing liver disease, rapid progression of metabolic disorders or the presence of considerable necrosis on liver biopsy. These patients should undergo OLT with a lower threshold [72].

#### 3.4.8. Investigational Drugs

In vitro studies demonstrated that AZD1480, an ATP-competitive inhibitor of JAK1 (Janus kinase 1) and JAK2 (Janus kinase 2), could inhibit HAV genotype III replication in human hepatoma cells. AZD1480 demonstrated a reduction in the activity of HAV internal ribosomal entry site (IRES)-mediated translation of phoshorylated-STAT3 and La (a protein involved in RNA metabolism) [73].

Another in vitro investigation showed that zinc chloride suppressed HAV replication up to 62% in human hepatoma cells infected by HAV genotype III, with a better performance if zinc was associated with interferon-alpha-2a compared to interferon alone [74].

The upregulation of enzyme heme oxygenase-1 through the administration of hemin, CoPP-9 (organic porphyrin) or andrographolide can suppress HAV replication without cell toxicity [75].

**Table 1 viruses-15-01080-t001:** Drugs utilised (or evaluated) for the treatment of acute HAV.

Drug	Results	Reference
N-acetylcysteine	Ineffective	Gunduz et al. [53]
Interferon-alpha	In vitro inhibition of viral replication Four patients showed improvement in liver function	Crance et al. [54], Yoshiba et al. [55]
Sofosbuvir	“In vitro” HAV RNA decrease	Wang et al. [50]
Corticosteroids	Survival improvement in children with fulminant hepatitis Reduction in the risk of prolonged cholestasis	Zakaria et al. [59], Yoon et al. [69], Daghman et al. [70], Jayappa et al. [38]
AZD 1480	In vitro reduction in viral replication	Jiang et al. [73]
Zinc Chloride	In vitro reduction in viral replication	Kanda et al. [74]
Heme oxygenase-1	In vitro reduction in viral replication	Kim et al. [75]

## 4. HEV Infection Overview

Hepatitis E is caused by being infected with the hepatitis E virus (HEV), the most recently discovered of the currently known hepatotropic viruses. HEV was discovered between the 1950s and the 1980s, during a large outbreak of unexplained acute hepatitis in India (Delhi, 1955–1956; Kashmir Valley, 1978), where the affected patients lacked serological markers of both hepatitis A and B infections [76]. These outbreaks were retrospectively confirmed as being caused by HEV, and they were the first observations of the excess maternal mortality associated with HEV [77]. Initially, it was referred to as enterically transmitted non-A, non-B hepatitis because the first identified transmission route was the faecal–oral one. The viral particles were not identified until 1983, when the Russian virologist Balayan and colleagues visualised the virus by electron microscopy while examining a sample of their own faeces after the ingestion of a pooled faecal extract of infected soldiers [78]. This was followed by the sequencing and cloning of the viral genome, and the identified agent was named HEV, as it was the fifth major hepatotropic virus to be recognised [79].

HEV is a small particle (27–34 nm in diameter), consisting of an icosahedral protein capsid that contains a single-stranded, positive-sense RNA genome. The virions are non-enveloped in the bile and faeces, while they are coated in a lipid membrane (quasi-enveloped) in the bloodstream so that they can be protected from inactivation by circulating specific antibodies [80].

HEV belongs to the *Hepeviridae* family, with two sub-families (*Parahepevirinae* and *Orthohepevirinae*); the *Orthohepevirinae* includes four genera (*Rocahepevirus*, *Paslahepevirus*, *Chirohepevirus* and *Avihepevirus*). The genus *Paslahepevirus* comprehends eight genotypes (GT1–GT8). GT1 and GT2 seem to infect only humans, GT3 to GT6 other mammals, and GT7 and GT8 infect camels [76,77,81]. The HEV genome consists of three open reading frames (ORFs): ORF1 includes 1693 codons, and it codes for proteins, such as methyl-transferase, RNA helicase, RNA polymerase and cysteine protease, which are responsible for the processing and viral replication [82]. ORF2 codes for viral capsid protein, and the 123 codons ORF3 encode for a viral porin involved in the release of infectious virions from infected cells [83].

As mentioned before, among the eight distinct HEV genotypes, only HEV-1, HEV-2, HEV-3 and HEV-4 are able to infect humans, although rare cases of HEV-7 have also been reported in the Mediterranean region [84,85,86,87]. HEV-1 and HEV-2 are obligated human pathogens and are spread, in the context of epidemic outbreaks, by the faecal–oral route via contaminated water. HEV-3 and HEV-4 are transmitted from animals, especially pigs, and less frequently from boars and deer [88]. They cause 3 million symptomatic cases, including 56,000 fatal courses, each year [89]. Infections by different genotypes occur in distinct epidemiological patterns and genotype distribution. HEV hepatitis cases are caused by genotypes 1 and 2 in Asia (mainly genotype 1), Africa (both genotypes 1 and 2) and Mexico (mainly genotype 2). In these settings, fragile sanitary infrastructure leads to inadvertent faecal contamination of water supplies (especially after heavy rainfall and flooding), resulting in inter-human transmission via the faecal–oral route [90]. Conversely, in high-income countries, HEV is usually acquired with food, in particular through pork products or wild boar meat [91]. HEV is often detected in pigs’ liver, where the concentrations of RNA are higher, but an infectious form of HEV can be found in other splanchnic organs and in pigs’ muscles. The ingestion of meat products from infected animals may result in animal-to-human transmission [88]. The faecal excretion in pigs is responsible for environmental contamination that leads to the maintenance of infection in pigs [92]. Boars are another animal HEV reservoir and are mostly infected by HEV-3, although HEV-4 is occasionally found [93]. The route of transmission is often related to the consumption of their meat. Therefore, the most frequent route of transmission is the faecal–oral route, but some cases of parenteral transmission after blood transfusion have been reported [94,95]. Vertical transmission, while rare, is possible and may lead to serious disease and possibly death in newborns [96,97].

Dual infection HAV/HEV is possible because the two viruses share a common transmission route (faecal–oral) and seems to be correlated with the presence of contaminated water that can be directly ingested or used for irrigation. The incidence of dual infection varies among different studies, as well as the outcomes: some authors suggest that there is no difference in mortality and severity of infection between hepatitis induced by a single virus compared to dual infection, while others report worse outcomes [98,99,100,101,102,103,104,105,106,107,108]. Dual infection may be more prevalent in certain subpopulations during HAV outbreaks. One study has indeed shown a higher presence of anti-HAV and anti-HEV in men who have sex with men [109]. These conflicting data may be due to geographical biases, risk factors and viral genotypes, so further studies are needed to evaluate the prevalence and outcomes of dual infection [110].

### 4.1. Epidemiology

According to WHO, there are 20 million HEV infections worldwide every year, causing an estimated 3.4 million symptomatic hepatitis E cases. However, a more recent study showed that nearly 19.44 million hepatitis E cases occur annually worldwide [111]. As stated by a recent meta-analysis about the seroprevalence rates, Poland, Denmark and France have the highest reported prevalence [112], while a seroprevalence and seroincidence study revealed that, in Germany, there are more than 400,000 infections per year [113]. Recently, several studies demonstrated a ‘hotpoint’ distribution, as most cases occurred in France, Scotland, central Italy, western Germany and western/central Poland. This led the European Centre for Disease Prevention and Control (ECDC) to establish a ‘HEV net’ with the aim of collecting HEV sequences of both humans and animals in order to better understand its distribution and epidemiology [77]. The European infected population is often represented by elderly patients, and it is estimated to have an incidence of 2 million new cases every year [114]. In Europe, nearly all infections are caused by HEV-3 [77]. In addition, the infection rate has increased significantly over the years due to the improvement in the detection methods and the testing frequency [115].

### 4.2. Pathophysiology

The entry of HEV in the host cells triggers the host’s innate immunity to produce a vigorous interferon response against infected hepatocytes [116]. This mechanism, together with CD4 and CD8 T-cell response, typically leads to viral clearance, especially in acute E hepatitis. However, in some special populations, such as immunocompromised patients where CD4 and CD8 T-cell activity is significantly reduced, the infection may persist, leading to chronic infection [117]. There is evidence that HEV infection may not be limited to the liver, as HEV RNA has been detected in the brain, kidney and placenta [118]. A long-time immunity against HEV infection persists after recovery or vaccination, and no symptomatic hepatitis was noted in reinfected patients [119,120].

### 4.3. Clinical and Laboratory Manifestations

HEV-1 and HEV-2 usually affect young individuals (between 15 and 30 years old), especially in low-income countries. The mean incubation period is 6 weeks and about 20% of people exposed to HEV-1 and HEV-2 manifest symptoms [89,121]. In a study conducted by Mansuy et al., among patients with HEV who develop clinical manifestation, 95% experience a self-limiting disease [122], characterised by fever in 27% of cases, jaundice (60%), asthenia (40%), nausea (10%), abdominal pain (11%), malaise, decreased food intake (8%), hepatomegaly and diarrhoea (5%) [123]. A subset of patients, between 0.5% and 13%, develop acute liver failure (ALF) and fulminant hepatitis, while up to 30% of patients could manifest acute icteric hepatitis [124,125]. Cases of extrahepatic manifestations were described and mostly consist in neurological, renal and haematological involvement. Neurological manifestations include neuralgic amyotrophy, Guillain–Barrè syndrome, encephalitis and Bell’s palsy [126]. Renal involvement is often manifested as membranoproliferative glomerulonephritis or membranous glomerulonephritis with or without cryoglobulinemia [127]. In Table 2, the most common extrahepatic manifestations are reported. HEV-1 and -2 are not known to be responsible for chronic hepatitis, while fulminant hepatitis has only been described for genotype 1. A study evaluating the kinetics of HEV antigen (HEV Ag), anti-HEV IgM and HEV RNA during infections was conducted on 24 sera from patients with acute hepatitis due to HEV-1. The study revealed intriguing data: high levels of HEV Ag correlate with a higher likelihood of fulminant hepatitis; HEV Ag and HEV RNA levels are no longer detectable after 4–8 weeks in patients who recovered; and in patients who develop fulminant hepatitis, the IgM titre is generally higher [128]. HEV RNA can be identified in stools 3 to 5 days before the onset of jaundice, and it disappears 2 to 3 weeks later [129]. The development of jaundice is usually followed by a marked increase in serum transaminase levels up to 10 times the normal limit [76]. As in HAV infection, a small proportion of patients can develop a cholestatic form characterised by jaundice and pruritus, which last for weeks and usually vanish spontaneously [76]. Mortality ranges from 0.2% to 4%, but it can reach a higher percentage in some categories [130]. In fact, women during the second and third trimester of pregnancy have 25% mortality, and in subjects with pre-existing liver disease, it is correlated with a reported mortality of 0 to 70% [131].

HEV-3 is asymptomatic in approximately 95% of the cases, while the other 5% develop elevation of liver enzymes, jaundice, itchiness, fatigue, anorexia and ALF in rare cases [77,132]. There are few studies investigating the outcomes of infection by HEV-3 or HEV-4 in pregnant women, and their results do not suggest a correlation between fulminant hepatitis and pregnancy [133,134,135,136,137]. The persistence of HEV viremia for six months defines a chronic HEV infection. Among patients that received solid organ transplantation and were infected with HEV, chronic HEV infection developed in around 60% of patients [138,139]. Case reports and case series of chronic HEV infection in immunocompromised patients, either by the illness itself or by pharmacological therapy, have been published [140,141]. A case report of an HIV-infected patient who developed HEV-related liver cirrhosis despite recovery of the immune system was described by Ingiliz [142]. In a study conducted by Kamar et al. on transplanted patients with HEV infection, 32% of patients were symptomatic for fatigue (24%), diarrhoea (6%), arthralgia (5%), abdominal pain (2%), jaundice (1%), fever and nausea (1%) and a significant rise in transaminases and the cholestasis index were observed [143].

### 4.4. HEV Treatment

Therapeutic options differ between acute and chronic infections. In most cases of acute hepatitis E, the infection resolves spontaneously, and no specific drugs are licenced or needed for the disease treatment. However, as mentioned before, some acute hepatitis E may evolve into liver failure. Chronic hepatitis usually affects immunosuppressed patients as solid-organ-transplanted patients and, in this setting, pharmacological treatments are necessary. Table 3 shows a brief summary of the drugs utilised (or evaluated) for the treatment of chronic HEV infection.

#### 4.4.1. Ribavirin

Ribavirin is a guanosine analogue that is phosphorylated intracellularly. This phosphorylation creates monophosphate, diphosphate and triphosphate forms of ribavirin. The triphosphate form is incorporated in viral RNA during RNA synthesis conducted by the RNA polymerases causing the formation of truncated RNA. With this mechanism, ribavirin can suppress viral replication [144]. The monophosphate form of ribavirin acts on inosine monophosphate dehydrogenase, causing the intracellular reduction in guanosine triphosphate that is fundamental for viral replication [145]. Ultimately, another mechanism of action was proposed. Ribavirin supposedly increases the rate of viral genome mutations, which, along with the limited reliability of RNA polymerase, leads to catastrophic errors in genome replication [146]. All of these mechanisms were studied in HCV infection.

The European Association for the Study of the Liver (EASL) stated that ribavirin therapy leads to liver enzyme normalisation and the eradication of HEV RNA in acute HEV infection [77]. However, this statement is based only on a retrospective study that enrolled 21 patients [147]. In these case reports, two patients with severe presentation of acute HEV hepatitis were treated with ribavirin and showed good recovery and clearance of HEV RNA at one month. Another case of acute HEV hepatitis was reported by Gerolami et al., and it was treated with ribavirin 600 mg twice daily, leading to a rapid improvement of liver function tests and a decrease in viraemia [148]. Pischke et al. successfully treated one case of acute HEV hepatitis genotype 1 in a 42-year-old woman with ribavirin [149]. At present, there is a lack of information on the use of ribavirin in acute hepatitis E in immunocompetent patients. Further studies must be conducted to assess the dose, duration and benefit in this setting.

The use of ribavirin in chronic HEV hepatitis is more extensively studied, especially in immunocompromised patients. In a case report described by Mallet et al., two immunocompromised patients, one due to a kidney and pancreas transplant and the other one due to idiopathic CD4+ T lymphocytopenia, were successfully treated with ribavirin [150]. A pilot study conducted by Kamar et al. on kidney-transplanted patients with chronic HEV infection treated with ribavirin for 3 months showed a sustained virological response in 66% of patients (four patients out of six) and relapse in two patients within 2 months [151]. A retrospective multicentre study involving 59 patients who had received a solid organ transplant with chronic HEV infection and were treated with ribavirin 600 mg/daily for 3 months showed HEV clearance in 95% of patients, but a relapse in 17% of these patients [152]. A large European retrospective multicentre study conducted on solid organ transplant recipients with chronic HEV infection receiving ribavirin at the median dose of 600 mg/daily showed a sustained virological response in 81.2% of patients after 3 months [153]. Mulder et al. conducted a retrospective multicentre cohort study of 92 adult transplant recipients with chronic HEV infection to determine the therapeutic range for ribavirin and found an optimal range between 1.8 and 2.3 mg/L, obtaining a sustained virological response after 3 months of therapy in 60% of all patients [154]. Friebus-Kardash et al. used a ribavirin regimen in 12 kidney-transplanted patients and achieved a sustained virological response in 94% of patients without decreasing the immunosuppressive regimen, although a significant reduction in haemoglobin levels was noted [155]. Ribavirin use is associated with the occurrence of anaemia and, less frequently, with skin reaction and dry cough. As stated by EASL Clinical Practice Guideline in immunosuppressed patients with chronic HEV infection, ribavirin is suggested for those who do not clear HEV RNA after a reduction in immunosuppressive therapy [77]. Ribavirin treatment is contraindicated in pregnant patients due to teratogenic potential. Several ribavirin treatment failures due to the resistant phenotype of HEV were reported; the more frequent mutation found was G1634R, but Y1320H, K383, D1384G, V1479I and Y1587F were also identified in ribavirin failure [156,157,158]. The reported ribavirin treatment failures may possibly be due to the type of immunosuppressive therapy, as a cases series of 12 patients in Asia showed an increased number of HEV hepatitis relapses in kidney transplant recipients probably as a result of a higher immunosuppressive regimen and adverse effects [159].

#### 4.4.2. Pegylated Interferon-Alfa

Interferon-alfa (INF-α) belongs to a family of proteins normally produced by the immune system with a wide range of biological effects, including antiviral activity, immunomodulatory activity, regulation of cell differentiation and inhibition of angiogenesis. When recombinant IFN is bonded to polyethylene glycol molecules, it forms pegylated interferon-alfa (PegINF-α): this modification leads to better pharmacokinetic characteristics, lengthening the pharmacological duration. After binding with this extracellular receptor, an enzymatic cascade involving the JAK/STAT system results in an increased expression of the genes coding for 2′-5′-oligoadenylate synthetase and the Mx protein homolog. The first one activates an RNase L that cleaves viral RNA [160], while the MX protein prevents viral transcription by inhibitions of trafficking and the activity of viral polymerases. Another mechanism involving dsRNA-dependent kinase (PKR) was described. Cells express dsRNA only if infected by viruses, so the stimulation of this mechanism could lead to a reduction in protein synthesis selectively in infected cells [161].

Since the therapeutic effects of interferon-α are based on the stimulation of the immune system, PegINF-α has been used to treat chronic HEV infection only in liver-transplanted or HIV patients, while other solid-organ-transplanted patients were not treated with INF-α due to the high risk of rejection [77]. There are limited data on the safety and efficacy of interferon in HEV-related hepatitis, and therefore, there is no universal recommendation for its use. A systematic review of antiviral therapy of chronic hepatitis E in immunocompromised patients showed that a sustained virological response was reached in six out of eight patients in the first three months of follow-up, while data about persistence of response at 6 months or longer were only available for two patients [162,163,164,165,166,167]. A report by Kamar et al. regarding one patient that received a liver transplant and was treated successfully with a PegINF-α regimen for three months for chronic HEV demonstrated viral clearance achieved after 3 months [168].

#### 4.4.3. Sofosbuvir

As much as 20% of patients with chronic hepatitis E do not respond to ribavirin, either for the selection of viral variants or due to discontinuation caused by adverse effects. Consequently, sofosbuvir was tested as an alternative approach. Sofosbuvir is a prodrug that is triphosphorylated inside the cells and acts as a uridine nucleotide analogue. Sofosbuvir blocks NS5B, a non-structural protein fundamental for HCV replication, by determining an RNA chain termination [169,170]. Sofosbuvir regimens were studied in six case studies, but the results were inconclusive because three reported a failure and the other three a success in the clearance of HEV together with ribavirin [171]. In 2020, the SofE, a pilot study evaluating the antiviral efficacy and safety of sofosbuvir in monotherapy, was concluded. After 24 weeks of treatment, HEV RNA was not reduced significantly [172]. Fraga et al. reported a case study in which a cirrhotic liver transplant recipient was treated with sofosbuvir unsuccessfully; HEV RNA was still detectable during the sofosbuvir regimen [173]. Further studies evaluating sofosbuvir in association with other drugs should be conducted.

#### 4.4.4. Vaccine

A vaccine containing amino-acid sequences of the ORF2 capsid protein of HEV genotype 1 was developed and licenced in China in 2010 (the commercial name is Hecolin^®^) [174]. A phase III clinical trial involving more than 100,000 healthy subjects demonstrated an efficacy higher than 99% [175]. Other trials evaluated the safety, protective effect and immunogenicity of the vaccine (NCT03168412 and NCT02417597). All of these trials were Chinese and demonstrated a cross-genotype action because HEV genotype 4 is at present the most frequent genotype in China [176]. Another study evaluated the persistence of seropositivity after vaccination with Hecolin^®^, predicting that, with the use of the power-law model and power-law modified model, 82.1–99.4% of the participants would remain seropositive for anti-HEV IgG for 30 years after vaccination [177]. The persistence of seropositivity for HEV was noted 4.5 years after vaccination in previous seronegative patients [178]. NCT03827395 is a phase Ia/Ib trial conducted in the USA that enrolled 25 healthy patients and non-pregnant females to test the safety, reactogenicity and immunogenicity of HEV-239, a 239 amino-acid subfragment of Hecolin^®^. On day 29, after the second dose, 100% of the subjects had a fourfold rise in serum hepatitis-E-virus immunoglobulin G concentration (NCT03827395). Hecolin^®^ was tested in women of childbearing age in rural Bangladesh in the NCT02759991 trial. In this phase 4 study involving 20.745 non-pregnant women, maternal and neonatal deaths caused by HEV were reduced by immunisation with HEV p239 [179]. There are two other candidate vaccines evaluated in clinical trials: both are based on virus-like particles [180]. The first one is the p495-based vaccine, and it was prepared from insect cells. The efficacy, after a complete vaccination cycle, was 95%, but the study was stopped in phase II due to a lack of commercial value [181]. The second one, a p179 HEV genotype 4-based vaccine, obtained from *E. coli*, was tested in a phase I clinical trial, resulting in good safety and tolerance [182]. The p179-based vaccine demonstrated a cross-genotype action against HEV genotypes 1 and 4 in Rhesus monkeys [180,182].

**Table 3 viruses-15-01080-t003:** Drugs utilised (or evaluated) for the treatment of chronic HEV infection.

Drug	Result	Reference
Ribavirin	Reduction in HEV viral replication, especially in immunosuppressed patients in acute and chronic HEV-related hepatitis	Gerolami et al. [148] Pischke et al. [149] Mallet et al. [150] Kamar et al. [151] Kamar et al. [152] Kamar et al. [153] Mulder et al. [154] Friebus-Kardash et al. [155]
PegINF-a	Tested in liver-transplanted patients, but also in HIV patients. Showed a sustained virological response in about 70% patients after 6 months. Used in 2 cases of kidney transplant.	Peters van Ton et al. [162] Haagsma et al. [163] Kamar et al. [164] Alric et al. [165] Singh et al. [166] Kamar et al. [167] Kamar et al. [168]
Sofosbuvir	Non-significant HEV RNA reduction after 24 months of treatment Failure in HEV clearance	Cornberg et al. [172] Fraga et al. [173]

## 5. Conclusions

Although usually benign and self-limiting, HAV and HEV infections have the potential to cause significant morbidity and mortality, particularly in some special populations such as immunocompromised patients, pregnant women or chronic-liver-disease patients. Outbreaks often burden low-income countries, possibly because of poor hygiene conditions, but high-income countries are nonetheless affected. Prevention strategies are key, and they should be implemented to reduce the probability of outbreaks. Among them, the presently available vaccines for HAV and the investigational vaccines for HEV have a pivotal role. Therapeutic options in non-self-limited cases or high-risk patients are limited and include both repurposed agents with antiviral effects (such as interferon and sofosbuvir in HEV and HAV, and ribavirin in HEV) and supportive care altering the pathogenesis of the disease (such as corticosteroids in HAV). In addition, some investigational agents in development have shown in vitro activity against HAV. To date, pathogen-specific aetiological therapy is far from being established both for HAV and HEV, and further research is needed.

## Figures and Tables

**Figure 1 viruses-15-01080-f001:**
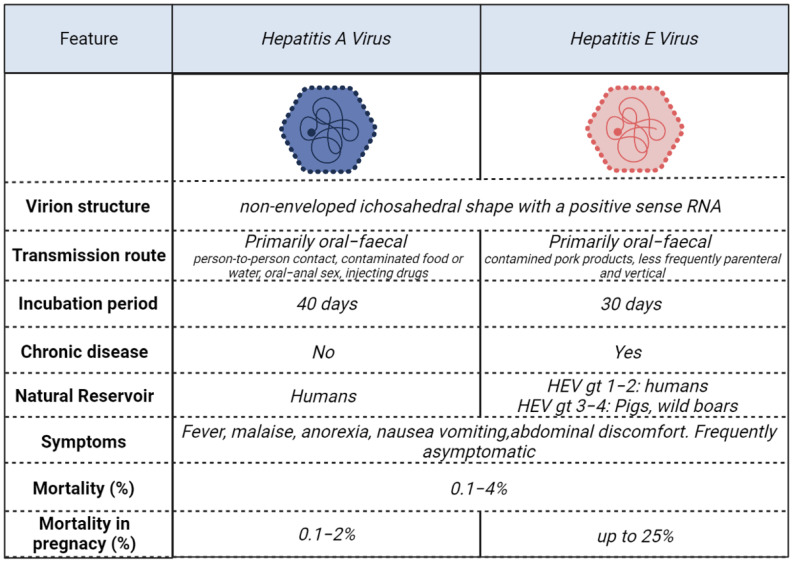
Differences and similarities between HAV and HEV. gt = genotype.

**Figure 2 viruses-15-01080-f002:**
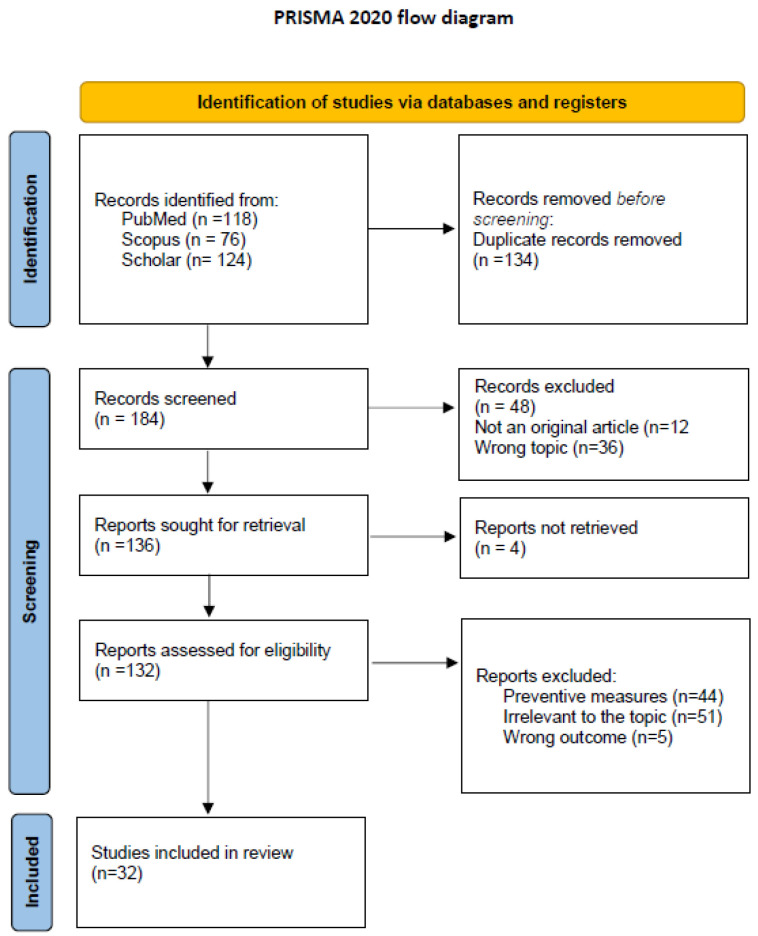
Study selection process for HAV. PRISMA 2020 flow diagram [1]. For more information, http://www.prisma-statement.org/ (accessed on 30 January 2023).

**Figure 3 viruses-15-01080-f003:**
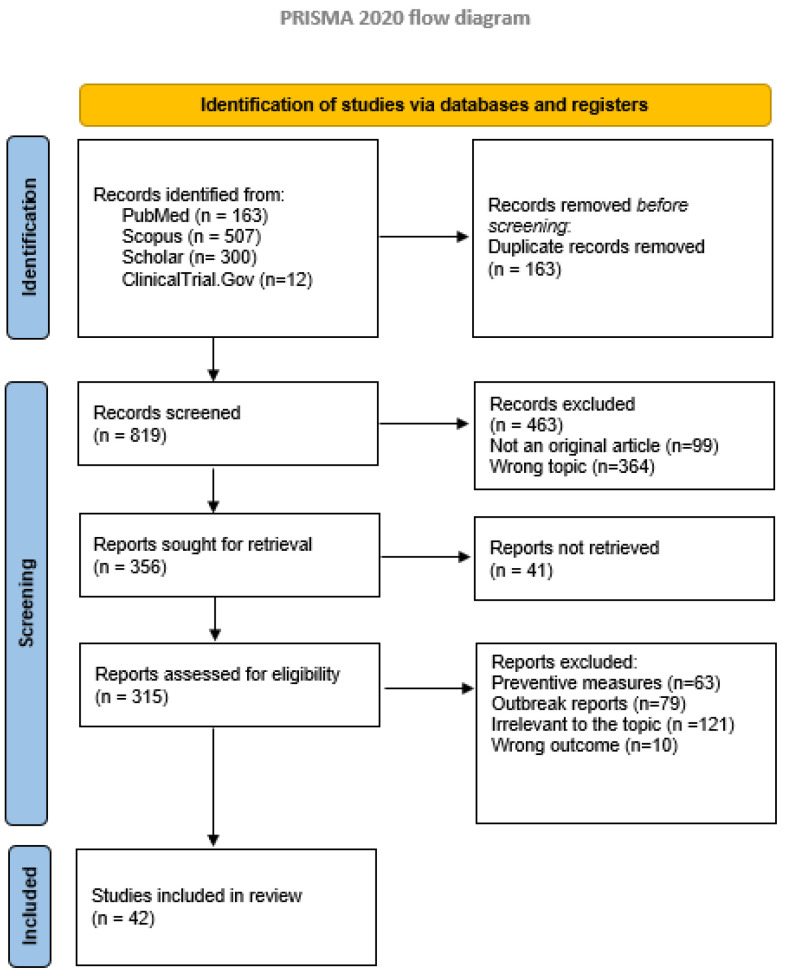
Study selection process for HEV. PRISMA 2020 flow diagram [1]. For more information, http://www.prisma-statement.org/ (accessed on 30 January 2023).

**Table 2 viruses-15-01080-t002:** Extrahepatic manifestations in HEV infections.

Organ/System Involved	Manifestation
**Neurological**	Bell’s palsy Encephalitis Guillain–Barré syndrome Myositis Neuralgic amyotrophy Oculomotor palsy Polyradiculoneuropathy Seizure Vestibular neuritis
**Cardiological**	Myocarditis
**Pancreas**	Pancreatitis
**Kidney**	Membranoproliferative glomerulonephritis Cryoglobulinemia
**Bone**	Polyarthritis
**Vascular**	Henoch–Schönlein purpura
**Haematological**	Aplastic anaemia Thrombocytopenia Haemolytic anaemia Aplastic anaemia Hemophagocytic syndrome CD30 (+) cutaneous T-cell lymphoproliferative disorder Thrombotic thrombocytopenic purpura Monoclonal gammopathy of uncertain significance (MGUS)

## Data Availability

No new data were created.

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
