# Peer review of "Treatment Options for Hepatitis A and E: A Non-Systematic Review"

_viruses, 2023, doi:10.3390/v15051080_

Round 1
Reviewer 1 Report
Good non-systematic review of current literature
However this does not add value , authors can mention drugs which are approved for treatment , limitations of the use for Hepatitis A and E
Need to correct typo error in corticosteroid dose in Hepatitis A- ots mg/kg/day not "die"
Author Response
Good non-systematic review of current literature
However this does not add value , authors can mention drugs which are approved for treatment , limitations of the use for Hepatitis A and E
Need to correct typo error in corticosteroid dose in Hepatitis A- ots mg/kg/day not "die"
Answer
Thank you for your excellent observations. The errors have been corrected and the document has undergone extensive English proofreading
Please see the attachment to read the new version.

Reviewer 2 Report
The manuscript aims to present the newest data on treating hepatitis A and hepatitis E virus infections based on a non-systematic review of the literature of the last 42 years. These two infections still represent a significant problem, a public health problem in many parts of the world.
The paper is generally well-written and organized, with a short introduction to the field. Also, the methods are adequately presented, and the selection of the reviewed articles was presented using PRISMA diagrams. As the authors intended, the main aspects of this review regard the treatment of HAV and HEV infections. Therefore, the overview of those viruses may include the data presented at the beginning of each section into a subsection of epidemiology.
Regarding the improvements, I would recommend changing the title to emphasize the subject of the treatment, as this is the paper's primary focus.
The keywords may be better chosen. In the abstract, I would replace the sentence regarding the non-systematic review more early, and more data on the treatment must be included here.
Some editing errors should be corrected.
Author Response
The manuscript aims to present the newest data on treating hepatitis A and hepatitis E virus infections based on a non-systematic review of the literature of the last 42 years. These two infections still represent a significant problem, a public health problem in many parts of the world.
The paper is generally well-written and organized, with a short introduction to the field. Also, the methods are adequately presented, and the selection of the reviewed articles was presented using PRISMA diagrams. As the authors intended, the main aspects of this review regard the treatment of HAV and HEV infections. Therefore, the overview of those viruses may include the data presented at the beginning of each section into a subsection of epidemiology.
Regarding the improvements, I would recommend changing the title to emphasize the subject of the treatment, as this is the paper's primary focus.
The keywords may be better chosen. In the abstract, I would replace the sentence regarding the non-systematic review more early, and more data on the treatment must be included here.
Some editing errors should be corrected.
Answer
Thank you for your excellent observations. We added an epidemiology section to help the comprehension of the text. Regarding the title we change in: “Treatment Options for Hepatitis A and E: A Non-Systematic Review”.
we have taken into consideration the advice regarding keywords as you could see in the document and we inserted earlier the fact that it was a non-systematic review. We included more data on the treatment in the abstract section as you suggested.
The document has undergone extensive English proofreading and errors are fixed
Please see the attachment to read the new version

Reviewer 3 Report
In this review the authors discussed HAV and HEV as enterically transmitted viruses causing acute hepatitis. The authors discussed overview for each virus, pathophysiology, clinical manifestations, and treatment and vaccine option for each one.
However, I have some comments on the virus on the present form
a) This is not a systemic review, not new data generated by the authors. The authors did general discussion for each virus. Therefore, no need for methodology, figure 1 and figure 2.
b) The authors should discuss why they discuss HAV and HEV among viral hepatitis. Why not HAV and HBV for examples. Common and difference features for HEV and HAV should be addressed in the introduction.
c) For HAV and HEV: In general: the authors should take of reference citation, most the cited references from reviews not from the original articles. For examples EASL should not be references for HEV open reading frame description. Also, ref 80 is not related to HEV-7. Ref 81 can not be used to describe vertical HEV transmission. For HEV-1 infections and acute liver failure, please use the suitable references. The same for HAV.
d) HAV: outbreaks also reported in developed countries especially with imported food and men to men sex. This should be described.
e) section 4.2: most of the information of HEV clinical manifestation is not accurate or from old resources, especially from lines 373 to 383. For examples some studies showed that ALF could reach to 13% with genotype 1, viremia is present even in the symptomatic stage. Anti-HEV IgM, HEV Ag, and HEV RNA kinetics could differ from acute, recovery and FFH stage of genotype 1 infections. Also not all extrahepatic manifestations are listed.
f) For HAV and HEV classification are old. Please follow the recent classification for each virus.
g) What about HEV/HAV dual infections?
h) There are a lot of typo and English editing is highly recommended in the whole manuscript. For example. page 10 line 361: HEV-1 e HEV-2 usually affects:
Author Response
In this review the authors discussed HAV and HEV as enterically transmitted viruses causing acute hepatitis. The authors discussed overview for each virus, pathophysiology, clinical manifestations, and treatment and vaccine option for each one.
However, I have some comments on the virus on the present form
1) This is not a systemic review, not new data generated by the authors. The authors did general discussion for each virus. Therefore, no need for methodology, figure 1 and figure 2.
Answer: This article is a comprehensive review of the literature conducted in a non-systematic manner. Howewer, we have been expressly requested by the editor to follow the standards of a systematic review
2) The authors should discuss why they discuss HAV and HEV among viral hepatitis. Why not HAV and HBV for examples. Common and difference features for HEV and HAV should be addressed in the introduction.
Answer: The article will be part of a special issue on the treatments of viral hepatitis: sharing common features such as structural form and drug scarcity, both HAV and HEV will be discussed in a single article. We added a Figure 1 to show common and differences between HAE e HEV
3) For HAV and HEV: In general: the authors should take of reference citation, most the cited references from reviews not from the original articles. For examples EASL should not be references for HEV open reading frame description. Also, ref 80 is not related to HEV-7. Ref 81 can not be used to describe vertical HEV transmission. For HEV-1 infections and acute liver failure, please use the suitable references. The same for HAV.
Answer: Thank you for your recommendations. We fixed those mistakes
4) HAV: outbreaks also reported in developed countries especially with imported food and men to men sex. This should be described.
Answer: Thank you for your advices. We added it in section 3.1
5) section 4.2: most of the information of HEV clinical manifestation is not accurate or from old resources, especially from lines 373 to 383. For examples some studies showed that ALF could reach to 13% with genotype 1, viremia is present even in the symptomatic stage. Anti-HEV IgM, HEV Ag, and HEV RNA kinetics could differ from acute, recovery and FFH stage of genotype 1 infections. Also not all extrahepatic manifestations are listed.
Answer: We have modified section 4.2 according to your suggestions. We have added a part regarding the serology kinetics and included a summary table of extrahepatic manifestations of HEV.
6) For HAV and HEV classification are old. Please follow the recent classification for each virus.
Answer: Thank you for this observation. As you have indicated we modify the section following the latest ITCV classification
7) What about HEV/HAV dual infections?
Answer: We added a brief description about HEV/HAV dual infection at the end of section 4 (HEV infection overview)
8) There are a lot of typo and English editing is highly recommended in the whole manuscript. For example. page 10 line 361: HEV-1 e HEV-2 usually affects:
Answer: The document has undergone extensive English proofreading
Please see the attachment for read the new version

Round 2
Reviewer 3 Report
The authors addressed my suggestions
Author Response
thank you very much for the excellent information you have given us